# Design of a Lab-On-Chip for Cancer Cell Detection through Impedance and Photoelectrochemical Response Analysis

**DOI:** 10.3390/bios12060405

**Published:** 2022-06-13

**Authors:** Yu-Ping Hsiao, Arvind Mukundan, Wei-Chung Chen, Ming-Tsang Wu, Shang-Chin Hsieh, Hsiang-Chen Wang

**Affiliations:** 1Department of Dermatology, Chung Shan Medical University Hospital, No.110, Sec. 1, Jianguo N. Rd., South District, Taichung City 40201, Taiwan; cshy713@csh.org.tw; 2Institute of Medicine, School of Medicine, Chung Shan Medical University, No.110, Sec. 1, Jianguo N. Rd., South District, Taichung City 40201, Taiwan; 3Department of Mechanical Engineering, Advanced Institute of Manufacturing with High Tech Innovations (AIM-HI), Center for Innovative Research on Aging Society (CIRAS), National Chung Cheng University, 168, University Rd., Min Hsiung, Chia Yi 62102, Taiwan; d09420003@ccu.edu.tw; 4Ph.D. Program in Environmental and Occupational Medicine, Kaohsiung Medical University, Kaohsiung 807377, Taiwan; u103803001@kmu.edu.tw (W.-C.C.); mingtswu@gmail.com (M.-T.W.); 5Research Center for Environmental Medicine, Kaohsiung Medical University, Kaohsiung 807377, Taiwan; 6Department of Public Health, Kaohsiung Medical University, Kaohsiung 807377, Taiwan; 7Department of Family Medicine, Kaohsiung Medical University Hospital, Kaohsiung Medical University, Kaohsiung 807377, Taiwan; 8Department of Plastic Surgery, Kaohsiung Armed Forces General Hospital, 2, Zhongzheng 1st Rd., Lingya District, Kaohsiung 80284, Taiwan

**Keywords:** lab-on-chip, linear interdigitated sawtooth electrode, dielectrophoretic impedance, photocurrent response measurement, electron beam evaporation, electrode lithography process

## Abstract

In this study, a biochip was fabricated using a light-absorbing layer of a silicon solar element combined with serrated, interdigitated electrodes and used to identify four different types of cancer cells: CE81T esophageal cancer, OE21 esophageal cancer, A549 lung adenocarcinoma, and TSGH-8301 bladder cancer cells. A string of pearls was formed from dielectrophoretic aggregated cancer cells because of the serrated interdigitated electrodes. Thus, cancer cells were identified in different parts, and electron–hole pairs were separated by photo-excited carriers through the light-absorbing layer of the solar element. The concentration catalysis mechanism of GSH and GSSG was used to conduct photocurrent response and identification, which provides the fast, label-free measurement of cancer cells. The total time taken for this analysis was 13 min. Changes in the impedance value and photocurrent response of each cancer cell were linearly related to the number of cells, and the slope of the admittance value was used to distinguish the location of the cancerous lesion, the slope of the photocurrent response, and the severity of the cancerous lesion. The results show that the number of cancerous cells was directly proportional to the admittance value and the photocurrent response for all four different types of cancer cells. Additionally, different types of cancer cells could easily be differentiated using the slope value of the photocurrent response and the admittance value.

## 1. Introduction

Cancer is one of the most prominent fatal diseases, with more than 200 types [1,2,3,4,5,6]. In 2020, approximately 19.3 million new cancer cases and 10 million cancer deaths were reported [7]. Cancer is a disease in which tissue cells proliferate abnormally to form tumors. The cells transfer to other parts of the body through the circulatory system and become circulating tumor cells. At the onset of the disease, these cells are extremely difficult to detect from blood or biopsy samples, and metastatic tumor cells show a unique pattern of activity [8].

Clinical studies show that the early detection of cancer can significantly increase the five-year survival rates of patients [9,10]. With the increase in air pollution rates, the cancer rate has been steadily increasing, and traditional techniques, such as magnetic resonance imaging, ultrasounds, and biopsies, are insufficient for early-stage cancer diagnosis [11], are expensive and time consuming, and sometimes generate false negatives [12,13,14]. Therefore, artificial intelligence (AI) cancer diagnosis methods have been built, which have been successful [6,15,16,17,18]. Despite modern scientific developments, the survival rates of the cancer patients remain low because of the failure of early-stage detection [10,19,20,21].

One way to detect cancer during its early stages is by using a cancer biosensor [22,23,24,25,26,27,28]. The specificity, quickness, compactness, and price of a biosensor present prospects for regionalized medical diagnoses [29]. Advances in the second generation of biosensors can be characterized by the application of antibodies or receptor proteins as molecular recognition elements, and signal converters are diverse, including field effect semiconductor devices, field effect semiconductors (FETs), optical fibers, piezoelectric crystal transistors, and surface acoustic wave devices (SAWs). The third generation of biosensors can be characterized by compactness, autonomy, and an instantaneous observation capability [30,31]. Microelectromechanical systems (MEMS) are used to fabricate array biosensors, but most of them are immunosensors. Enzyme array biosensors are unsuitable for covalent bonding or cross-linking technologies fixed on substrates because of the variability of enzymes.

Extensive research has been conducted on various biosensors used to diagnose cancer lesions [27,32,33,34,35,36,37,38]. A typical biosensor comprises a target cancer marker and a corresponding biotransducer, which is essential to the identification of the technical specifications of a device and a bioreceptor [39]. Biomarkers imply variations in the appearances or representations of proteins, and these features are associated with the evolution of a particular disease and its reaction to medication [40]. Hence, a biomarker can be a specific cell, molecule, gene, gene product, enzyme, or hormone [41,42]. To date, more than one hundred and sixty biomarkers have been introduced for cancer diagnosis. A lab-on-chip is a biosensor that facilitates the concurrent measurement of numerous biomarkers [43,44,45,46,47]. Advanced microfabrication techniques facilitate the integration of microfluidics with biosensing functionalities on a single sensor chip, allowing system automation.

Reduced glutathione (GSH) plays a crucial role in life forms, as it comprises a substantial amount of biological knowledge. A tiny divergence or an anomaly in the concentration of GSH is commonly correlated with numerous diseases, for example, an abnormality in GSH concentration in biological fluids or tissues is often directly associated with several medical diseases, including diabetes, cardiovascular diseases, and cancers [48,49,50,51,52,53,54,55,56,57,58,59]. The relative amount GSH and glutathione disulfide (GSSG) will differ between cancer lines [60]. However, many approaches such as resonance spectrometry, spectrofluorimetry, and colorimetry have low sensitivity, are time consuming, and involve long detection times [61,62,63,64,65].

In the last few years, numerous approaches have been studied to detect, identify, and characterize cancer cells. Of all the studied methods, electrical-impedance-measurement-based approaches provide numerous advantages such as high sensitivity, rapid detection, low cost, and suitability for integrated microsystems [66,67]. Differentiating cancerous cells on the basis of impedance measurements provides accessible knowledge on the characteristics of cancerous cells as a function of frequency [68]. Additionally, dilectrophoresis (DEP) is one of the most proficient methods which can be applied in the swift manipulation of bioparticles [69]. Recently, microfabrication technology has been chosen to produce microelectrode patterns that permit adequately substantial DEP forces to be produced onto cells with low applied voltages. In previous research, numerous infrequent cancer cells were efficiently exploited by DEP [70,71,72]. The impedance measurement with DEP in biosensors can be amalgamated to increase and magnify the sensitivity and lessen the time taken to detect cancerous cells [73]. This method is known as the dielectrophoretic impedance measurement method (DEPIM) [74]. Electrical impedance spectroscopy (EIS) is a label-free and non-invasive electrokinetic technique which, in recent years, has been used to detect and differentiate different types of cancer cells based on the cell polarization generated by an electric field and the interaction of ions along the cell surface [75,76,77].

Many recent articles have been conducted to differentiate and characterize the phases of cancerous cells. Many of these studies require complex surface marker labeling and have high levels of uncertainty, poor efficiencies, high costs, and long durations. Therefore, in this study, a biochip was fabricated using the light-absorbing layer of a silicon solar element combined with serrated interdigitated electrodes for the identification of CE81T esophageal cancer, OE21 esophageal cancer, A549 lung adenocarcinoma, and TSGH-8301 bladder cancer cells.

## 2. Materials and Methods

### 2.1. Design of Microelectrodes

The purpose of this experiment was to measure the impedance value of cells that accumulate between electrodes. Therefore, a linear interdigitated sawtooth electrode with an electrode gap was used to control and measure cells. The gap was circular and had a radius of 50 μm. The electrode was designed to measure the number of accumulated cells between electrodes and ensure that a sufficient number of cells can be generated when a high frequency voltage is applied. Moreover, it can ensure that a sufficiently large dielectrophoretic force can be generated on cells and is more sensitive than general microelectrodes.

### 2.2. Chip Fabrication

Monocrystalline silicon (mono c-Si) was used as the substrate for the manufacturing process. The mono c-Si substrate has a high photoelectric conversion efficiency. The standard yellow light lithography process was used for chip fabrication [78,79]. The process used the pattern defined by the photomask for substrate fabrication. The required electrode pattern was obtained, and the wafer-manufacturing process was divided into two steps: (A) Au/Cr electron beam evaporation and (B) the electrode lithography process.

#### 2.2.1. Electron Beam Evaporation of Au/Cr

Electron beam evaporation involves the bombardment of a target material in a vacuum environment for physical vapor deposition [80,81]. This method is used to convert the kinetic energy of a high-energy electron beam into heat energy to melt a target material. Coating is performed by using the saturated vapor pressure of a target material when it is close to its melting point.

#### 2.2.2. Electrode Lithography Process

S1813, a positive photoresist, was used in spin-coating. The silicon substrate was first cut to an appropriate size and treated with acetone, methanol, and DI water. The glass wafer was placed in a photoresist coater, and an appropriate amount of S1813 positive photoresist was added for coating. It could be uniformly spin-coated on the substrate at a height of approximately 3 μm through high-speed rotation. After the spin-coating of the photoresist, it was placed in a 90 °C hot plate for four minutes for soft baking. The purpose of this was to volatilize the organic solution in the photoresist through heating and prevent the sticking of the photoresist and photomask during exposure. The photoresist was flattened, and the adhesion between the photoresist and substrate was enhanced. Lithography can be divided into three types according to the light source: UV lithography, electron beam lithography, and X-ray lithography. This experiment used a general yellow light lithography process. High-pressure mercury and mercury–xenon arc lamps were in the UV wavelength range and had two high-intensity emission spectral lines in the UV wavelength range of 350–450 nm; that is, two high-intensity emission spectral lines were present, namely g-line (436 nm) and i-line (365 nm). The main purpose of their use was to make the photoresist bond or break in order for the components exposed to light to show considerable differences in the solubility of the developer and to accelerate the pattern transfer. Part of the light that was not absorbed by the photoresist during exposure reached the surface of the substrate through reflection. Incident light waves produced constructive and destructive interference, forming a standing wave effect, which subjected the photoresist to uneven light intensity and resulted in corrugation on the side of the photoresist and a change in the line width of the photoresist.

Post-exposure baking caused the rearrangement of the exposed photoresist and reduced the abnormality caused by the standing wave effect. In this experimental process, we baked the wafer at 90 °C for four minutes after exposure. The glass wafer was placed in a mixed MP351 developer and shaken for approximately 10–15 s. The special developer, MP351, of S1813 was used for development. After the photoresist layer detached, development was completed. Figure 1 shows the actual finished product of the biosensor chip.

### 2.3. Sample Preparation

Four types of cancer cells (CE81T, OE21, A549, and TSGH-8301) were used to prepare a culture medium according to the needs of the cells. The medium required for TSGH8301 cells was Roswell Park Memorial Institute–160 (RPMI-160), whereas the medium required for the CE81T, CE81T-4, and A549 cells was Dulbecco’s Modified Eagle Medium (DMEM) [82,83]. During preparation, 2.0 g of sodium bicarbonate (NaHCO_3_) was added to 1 L of RPMI-160. Hydrochloric Acid (HCl) or sodium hydroxide (NaOH) was added to balance the pH value between 7.1 and 7.3. Then, 1% (*v*/*v*) antibiotic–antimycotic and 10% (*v*/*v*) heat-inactivated fetal bovine serum (FBS) were added to inhibit the growth of bacteria or for cell growth. NaHCO_3_ (3.7 g/L) was added to DMEM. After sterilization, the glass jars were sealed and refrigerated. Finally, 1% (*v*/*v*) non-essential amino acids and 1% (*v*/*v*) antibiotic–antimycotic solution were added to the culture medium before use.

## 3. Results

### 3.1. DEP-Based Cell Concentration

Figure 2 shows the differences before and after the application of 10Vp-p AC voltage to the biosensor chip. In this experiment, 6000 CE81T and 30,000 CE81T cells were used as examples. A 1 MHz 10Vp-p sine wave was applied to the electrode for 10 min for DEP aggregation in a range of 3000–30,000. The cells exhibited positive dielectrophoretic forces. Figure 2a,c present the OM images of 6000 and 30,000 esophageal cancer cells before DEP was applied, respectively, and Figure 2b,d display the OM images of 6000 and 30,000 cells after DEP application, respectively. The cells aggregated in a bead shape near the tip of the microelectrode after DEP was administered.

### 3.2. Dielectrophoresis Impedance Spectroscopy Measurement

During the initial wafer measurement, no cells were injected above the wafer microelectrode. Only the impedance value of the sucrose solution was measured, and sucrose admittance was calculated. A different number of cells was used in each test to obtain the admittance values of the reference samples (see Appendix A for counting of the cells). Figure 3 shows the relationships among changes in the admittance values of the four different cancer cells in each test using a different number of cells. A linear relationship was found between the change in admittance value and the number of cells (see Appendix A for the calculation formula for cell admittance measurement). Changes in the relationships between the admittance value and the number of cells showed different slopes in cancer cells in different parts. The differences between the slopes can be used to identify cancer cells. As a non-invasive, label-free electrochemical method, impedance measurements can provide sensitive and quantitative measurements. These advantages mean impedance measurement methods are widely used in cell research and analysis, especially in live-cell analysis and long-term live-cell monitoring.

Figure 4 shows the photocurrent response measurement results for four cancer cells in tests using a varying number of cells. CE81T-1 and CE81T-4 were first- and fourth-stage lesions of esophageal cancer cells obtained from an East Asian population, whereas OE21-1 was the first-stage lesion of esophageal cancer cells obtained from a Caucasian population (see Appendix A for a schematic diagram of the reparation of the different cells). The photocurrent response curves of the cancer cells at different locations and degrees of cancerous lesions were measured with a microcurrent meter in each test using a different number of cancer cells. As shown in Figure 5, the relationship between the number of cells and the photocurrent response was linear, and the slopes of the photocurrent responses in cancer cells with different cancer lesions and the number of cells were quite different. Compared with the general optical measurement method, the cost was lower, and the detection process was relatively convenient and fast.

In a circumstantial view, our PEC sensor is indeed sensitive to GSSG. The ratio of GSH/GSSG, which affects the photocurrent response in the electron transport process, is a significant indicator for cancer detection. We can observe changes in photocurrent responses to measure different numbers of cancer cells, and our data, as shown in Figure 3 and Figure 4, could support this circumstantial view (see Appendix A for the photocurrent analysis of CE81T-1 and CE81T-4 cells). On the contrary, one of the limitations of this study could be the number of cancerous cells that the biosensor can detect. So far, we showed that the study could identify cancerous cells in numbers from 2500 cells to 30,000 cells. Even though the presence of 2500 cancerous cells is still considered to be early-stage cancer, a further study has to be conducted to detect cancerous cells from an earlier stage and improve the maximum limit of the detection. Limit of detection (LOD) analysis is one of the methods used to assess the quality of any biosensors [79]. The LOD analysis in this study showed that the mean LOD value was 3.095 cells. These values are considerably lower values, because less than 5000 cancerous cells indicate an early stage. However, an LOD of 3 is very low, which shows that this biosensor can detect extremely low numbers of cells (see Appendix A). The lower limit of quantification (LOQ) of the proposed biosensor was 9.378, which outstrips other detecting methods. Additionally, the average sensitivity of the biosensor was found to be 2.79 S/#m^−2^. The extremely good sensitivity, ease of manufacturing, and low cost of the proposed sensor suggests that this study can be analyzed for further to produce much more efficient biosensors in the future.

In Figure 3 and Figure 4, the slopes of photocurrent responses and admittance values of different cells stack up on each other. However, by combining both the measurements, as shown in Figure 5, the slopes of different cells are distinct. Therefore, it can be inferred that the sensitivity of the biosensor developed in this study can be increased by combining both the photocurrent response and admittance measurements. The manufacturing process of solar cells follows three steps, as shown in Figure 6. The first two steps in the process are similar to the manufacturing process of current mass-produced silicon solar cells. The first step is to use optical inspection methods to find any defects. The second step is cleaning and etching. The third step is phosphorus diffusion, which involves the use of a micro-galvanometer, which biases the wafer through a probe and uses a light source to excite the carriers to separate the sample. A bias voltage of 1v is given through a microcurrent meter, and an indoor light source is used to excite the sample. The light introduced will be absorbed by the material, thereby increasing the number of holes and electrons. As a result of this, the electric current will increase [84]. This phenomenon is known as photoconductivity, in which the electric current flowing is a result of the excitation caused by light striking a material [85]. Additionally, under light, photo-excited electrons and holes will be compelled to realign amid the two semiconductors due to the built-in electric field [86,87,88]. An indoor light source is easy to obtain, which meets the chip development expectations, and the distance between the light source and the chip can be fixed. By exiting the sample and creating a difference in potential across the junction of the two regions, a loss of electrons in the *n*-region and a subsequent gain in the p-region can be achieved. At this time, when cells are dropped on the N-type surface, the electrons will participate in a reduction reaction in GSSG, which will increase in conductance, because the total number of electrons concentrated on the N side is fixed, so the current will increase, thereby producing a photocurrent response.

## 4. Conclusions

Four different types of cancer cells, namely CE81T esophageal cancer, OE21 esophageal cancer, A549 lung adenocarcinoma, and TSGH-8301 bladder cancer cells were detected using a biochip manufactured by using a light-absorbing layer of a silicon solar element combined with serrated interdigitated electrodes. Dielectrophoretic impedance measurement with an interdigitated sawtooth microelectrode was used to generate a positive dielectrophoretic force on the electrode and generate a high electric field area to aggregate four cancer cells on the tip of the microelectrode. Then, a sine function was used for the measurement. The cells were aggregated using a dielectrophoretic force, the interdigital electrode gap had a large surface area that accommodated the cells, and the admittance value was obtained after calculation and then analyzed. To measure the photoelectric flow, a microcurrent meter was used to bias the wafer through the probe, and the photo-excited carriers were generated by the absorption of light through the element. The concentrations of GSH and GSSG were used to catalyze the separation of electrons and holes. The relationship between the degree of cancerous lesions and the photocurrent response was determined and used to detect different degrees of cancerous lesions in the same area. The relationship between the number of cells and the admittance value and the relationship between the number of cells and the photocurrent value were roughly linear. Cancer cells can be distinguished according to a difference in the slope, and thus, we can conclude that the dielectrophoretic impedance and photocurrent response measurement methods can be applied in clinical diagnoses.

## Figures and Tables

**Figure 1 biosensors-12-00405-f001:**
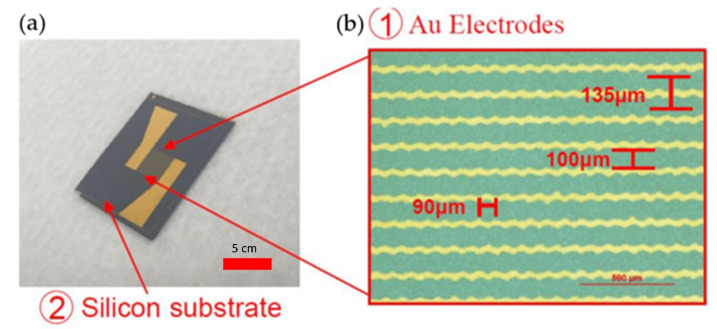
(**a**) Actual finished product of the biosensor chip. (**b**) Close view along with the dimensions of the electrodes.

**Figure 2 biosensors-12-00405-f002:**
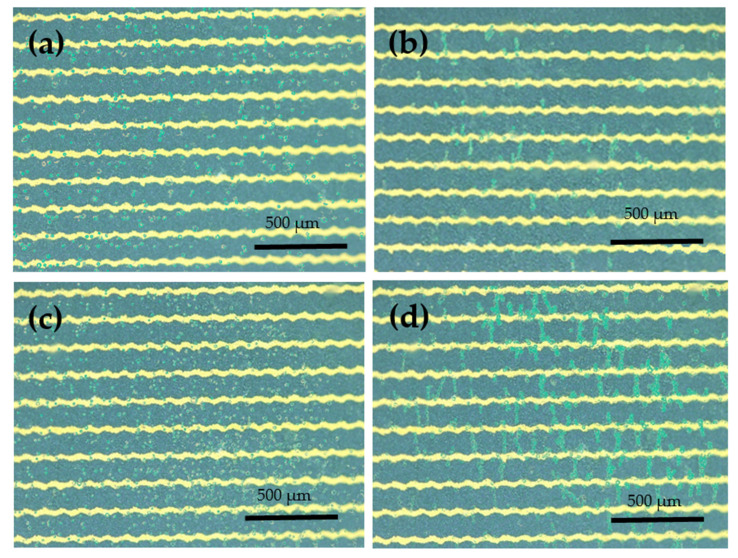
OM image before and after the application of 10Vp-p AC voltage to the biosensor chip. (**a**,**b**) Images of 6000 CE81T cells before and after the application of DEP, respectively. (**c**,**d**) Images of 30,000 CE81T cells before and after the application of DEP, respectively.

**Figure 3 biosensors-12-00405-f003:**
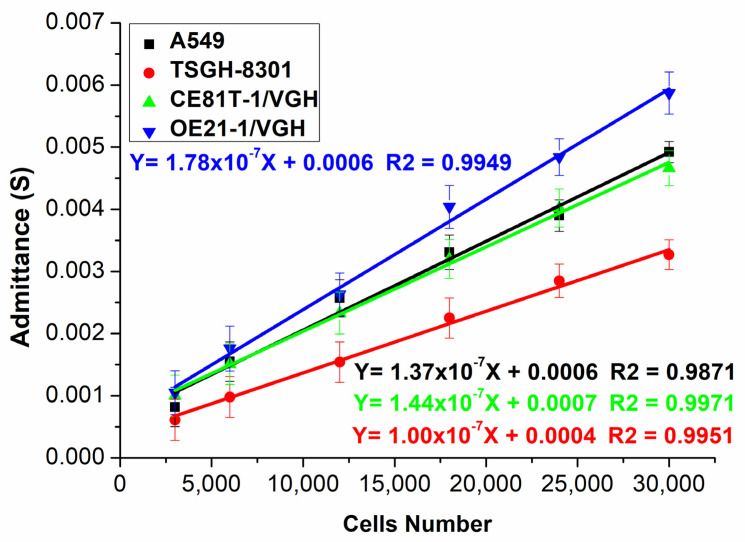
Admittance values vs. number of cells in four cancer cells. Error bars represent the standard deviation of the measurement (*n* = 5).

**Figure 4 biosensors-12-00405-f004:**
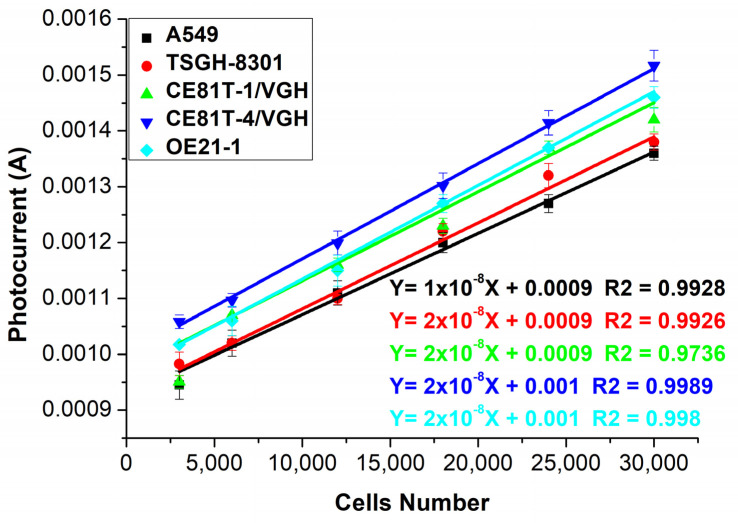
Photocurrent response measurement results of four cancer cells in tests using varying number of cells. Error bars represent the standard deviation of the measurement (*n* = 5).

**Figure 5 biosensors-12-00405-f005:**
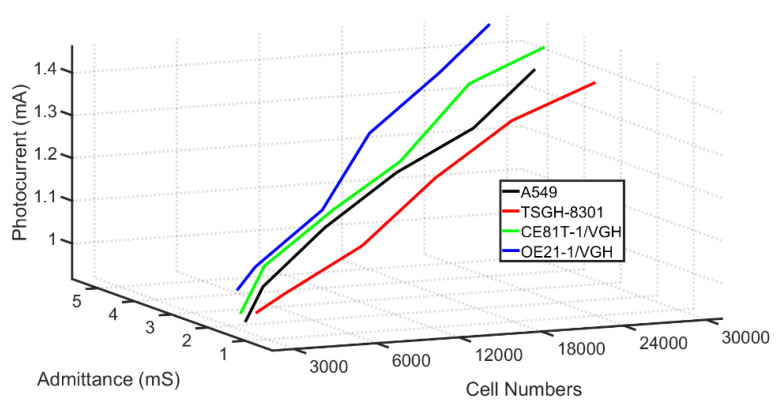
Three-dimensional plot of photocurrent response and admittance value measurement of the different cells.

**Figure 6 biosensors-12-00405-f006:**
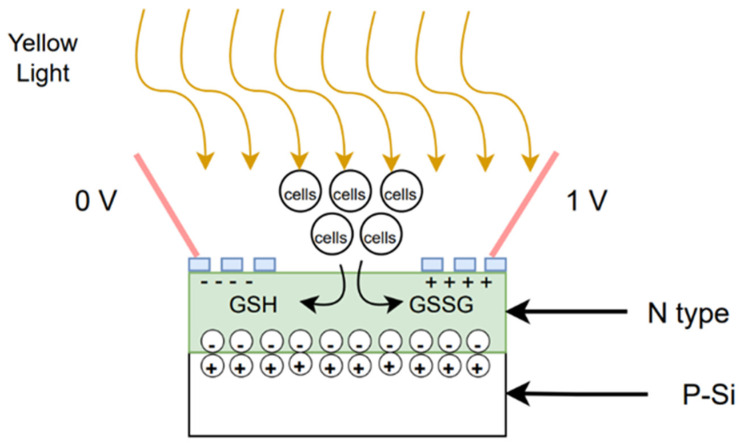
Schematic diagram of the biosensor developed.

## Data Availability

Data sharing not applicable.

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
