# Peer review of "Design of a Lab-On-Chip for Cancer Cell Detection through Impedance and Photoelectrochemical Response Analysis"

_biosensors, 2022, doi:10.3390/bios12060405_

Round 1

Reviewer 1 Report

In this version, authors have improved the quality of the manuscript, even there are still mistakes or unclear statements in these answers. The revised manuscript can be published in “Biosensors” after minor revision.

The authors should avoid the mistakes.

For example:

“The LOD analysis of this study shows that the shows that the mean LOD value is 3.095 cells.”

“Also, the average sensitivity of the biosensor was found to be 2.79 S/#m-2.”

……

Reviewer 2 Report

The manuscript “Design of a Lab-On-Chip for Cancer Cell Detection Through  Impedance and Photoelectrochemical Response Analysis” develops an electrode (chip) based system for cancer cell detection. The authors demonstrated meticulous characterizations of such chip system. They also demonstrate admittance and photocurrent’s correlation with four different cancer cell numbers. The work is interesting and precedent, although it is preliminary.

I think it is acceptable after some revision, taking into account the following points.

Major points:

  1. The system presented is novel, towards a cancer biochip sensor. Without specificity data, the clinical diagnosis application would be limited. Specifically, admittance or photocurrent vs cell number plot to a cell mixture (mixture of one type of cancer cell with normal cell; mixture of two types of cancers) would be important to present. Such data would demonstrate biochip sensor system’s specificity towards one type of cancer cell.
  2. Conclusion, Line 278 “The cancer cells can be distinguished according to difference in slope, and thus we can conclude that the die- lectrophoretic impedance and photocurrent response measurement methods can be ap- plied to clinical diagnosis.” This could be an overstatement, without specificity data. To distinguish the four cell lines presented, the authors need samples with different cell numbers. This would not be the case for a given sample in clinical settings, where a specific number of cells will be presented in the sample.
  3. Sample size is missing from data presentation. Figure 3 and 4 should fully describe data presentation with sufficient details (e.g., Mean ± SD). What is the sample size in Figures?
  4. Line 226. The authors reported a LOD. Please specify how LOD is mathematically calculated.
  5. The significant figures for the slope of the fitting curves need to be improved. ( Figure 3 and 4) For example, currently, in Figure 3, there are three curves with slope of 1E-07, which cannot be differentiated from each other. To improve, the unit of Y axis in Figure 3 could be mS, unit of Y axis in Figure 4 be mA or μA. Please present at least two significant figures for slopes in both figures.

Minor points:

  1. To help reader appreciate the scale of the images, it is beneficial to include scale bars in Figure 1a and Figure S6.
  2. Line 210. “Figure 4-3” could not be found in the manuscript.
  3. Line 282. The paragraph does not mention Figure S7, 8 , which are present in Supplementary Materials.
  4. In Supplementary Materials’ s Abstract, Section 5 is not mentioned. Only the first four sections are mentioned.

Reviewer 3 Report

The present manuscript reported “Design of a Lab-On-Chip for Cancer Cell Detection ThroughImpedance and Photoelectrochemical Response Analysis”. This paper has enough novelty, but, some parts need to be improved. Generally, what is the main purpose,? Why did you test four cancer cells? what different?

  1. page 4, 2.3 Sample preparation

Please read it again, line 168 and 172, repeated!!!

Some “growth of bacteria or molds and provide necessary nutrients” what do you mean?  It is only for stopping growing bacteria, 1% (v/v) of antibiotic–antimycotic, please modify it.

Line 175, is wrong! Then, 10% (v/v) heat inactivated FBS is added to inhibit bacterial or fungal growth. FBS is another thing.

  1. line196, please give more information about “admittance values”
  2. add more information about, impedance spectroscopy measurements, electrochemical parameters and etc.,
  3. please add more information about how did you count the cells in the paper.
  4. could you explain why the intensity signal OE21 is higher than TSGH cells?
  5. line 167, please explain, you added? how? “During preparation, 2.0 g/L sodium bicarbonate (NaHCO3) is added to RPMI-160.” How much the volume?
  6. line 218, please add more information on “different stages of the cancer cells” what stages?

Reviewer 4 Report

This topic has been popular in bioelectronics. The authors here proposed a novel mechanism to detect cancerous cells. I think this manuscript can be considered to publish after some revisions.

1. The authors should give characterization about the cells on chip during detection. Because the electromagnetic wave may have some influences on the cells, so the signal alteration from the cellular state should be excluded.

2. How about the detection results for the normal cells? I think the key point for this method lies in the selectivity. If possible, the authos should detect the mixed samples with normal cells and cancerous cells to see if the chip can differentiate them.

3. Did the authors measure the real sample from human or animals? The complex background is important for this method.

4. The authors should give error analysis. That is to say the auhors should repeat the experiments for many times.     

Round 2

Reviewer 2 Report

The manuscript “Design of a Lab-On-Chip for Cancer Cell Detection Through  Impedance and Photoelectrochemical Response Analysis” develops an electrode (chip) based system for cancer cell detection.

The authors have suitably edited the manuscript, addressing the reviewer comments and clarifying key elements. Specifically, combination of Photocurrent response and Admittance value for analysis drastically improved the sensor sensitivity.

I think it is acceptable after additional minor revision, taking into account the following points:

1.    Figure 3 and 4 should include clear statement about sample size in the figure legend. Use a prior Biosensors paper as an example, Figure 6 of the following paper https://www.mdpi.com/2079-6374/12/5/262 “Error bars represent the standard deviation of the measurement (n = 3)”. Such statement is important to have in the main text.

Author Response

The authors thank the reviewer for their positive comments and at the suggestion of “acceptance after additional minor revision”. The sample size used in this study was 5 for each case. As per the suggestions of the reviewer, the authors have added the following in the figures 3 and 4, “Error bars represent the standard deviation of the measurement (n = 5)”.

Reviewer 3 Report

it is improved.

Author Response

The authors thank the reviewer for their positive comments. It is due to their comments and suggestions that the quality of the article has improved drastically.

Reviewer 4 Report

I agree to publish this manuscript.

This manuscript is a resubmission of an earlier submission. The following is a list of the peer review reports and author responses from that submission.

Round 1

Reviewer 1 Report

In this paper, a lab-on-chip was fabricated based on the light-absorbing layer of a silicon solar element combined with serrated interdigitated electrodes for the application of detecting four types of cancer cells: CE81T esophageal cancer, OE21 esophageal cancer, A549 lung adenocarcinoma, and TSGH-8301 bladder cancer cells. The subject is worth of investigation, and the novelty is also acceptable. However, the experimental data are limited for the proposed lab-on-chip and need to be improved. My suggestion is that major revision is necessary before publishing in “Biosensors”. The detailed concerns that I have in the manuscript are listed as follow:

  1. Regarding to the ABSTRACT, the authors should provide the details information to readers achieved in the proposed work instead of just presenting as a generalization.
  2. The authors are advised to provide more details about impedance and photoelectrochemical response analysis strategies in the INTRODUCTION part because it is mainly presented in the TITLE of the manuscript. It is actually lack of the investigation of background of the strategies, including the introduction of established methods, the important achievements, and the relevant scientific interests.
  3. The author mentioned that the electron-hole pairs were separated by photo-excited carriers through the light-absorbing layer of the solar element. However, there is no explanation and conclusive evidence in the manuscript except for the limited illustration in the CONCLUTIONS part. It is advised that the authors should clearly show the mechanisms and the relative data to support it for readers. And the possible energy diagram for it is also advised to provide.
  4. The lab-on-chip was fabricated for detecting four types of cancer cells. However, the information about the performance evaluation of proposed lab-on-chip is limited. The authors should design experiments for investigating it, such as, limit of detections, selectivity, reproducibility, etc.
  5. What is the advantage of the proposed work? The authors are advised to provide a table to make comparison with the previous works.
  6.  What is the meaning of GSH and GSSG in the manuscript?

Reviewer 2 Report

Dear authors

The topic of this article is very interesting and actual and shows very promising and good results. 

However, some figures from the article should be meliorated as figures 4 and 5, the numbers of equations were cut. 

Best regards,

Reviewer 3 Report

This manuscript introduces on-chip cancer cell detection by both impedance and photoelectrochemical response. The scientific novelty is unclear. The two proposed methods show no benefits for sensitivity or accuracy improvements. The manuscript is lack of background information, critical experiments and analysis. It also lacks key information to support the results. I would suggest addressing the following concerns before further consideration.

Major comments:

  1. In abstract part, please add the statement for backgrounds and current issue to address. Then, summarize the authors’ idea and the proposed methods, and lastly point out key results and potential applications.
  2. The scientific novelty is not clear. There is not much innovation in the proposed impedance and photoelectrochemical sensing methods. What’s new and better?
  3. The introduction is lack of in-depth discussion of current issue for cancer cell detection. How does the proposed methods solve the problem.
  4. Figure 1, Please take a closer image for the biochip and add caption for the whole images 1 and 2. The image 1 cannot be seen clearly. The dimension labels of the electrodes were not matched with spacing. Please remark the dimensions. And I suggest use (a) and (b) to be consistent with the others.
  5. Figure 2, it is unnecessary to introduce cell culture as key results and to be put at Fig. 2, such an important position. Cell culture is usually viewed as fundamental experiment in biology.
  6. Lack of LOD analysis and discussion.
  7. Lack of selectivity test for the four types of cells. No supporting evidence the proposed methods can distinguish different types of the cancer cells.
  8. The authors seemed to present two combined methods. However, there seems no benefit for each proposed method since the two detection methods have already been reported in the literature. There is also no relation for the two proposed methods. There are not any advantages using the two combined methods than a single method either, for example better sensitivity or improved accuracy.

Minor comments:

  1. In the title, “lab-on-a-chip” is mostly seen and used.
  2. Figure 3, missing scale bars.
  3. Figure 4 and 5, the fitting functions were not clearly presented. Please check them.
  4. Line 53, I don’t understand what the authors were trying to convey by “however”

Round 2

Reviewer 1 Report

I can not find the answer for Q3. Actually, there are two section 2.1.1 in the updated manuscript which looks same as the old version. Besides, the authors should strictly think the Q4, and provide the relative investigations for readers in the manuscript. 

Reviewer 3 Report

The manuscript made several revisions, but still kept its weakness. Quality of presentation and scientific soundness are low. I would insist on rejection of this manuscript.

First of all, there is no meaning of detecting huge numbers of cancer cells (more than thousands) using cell lines for demonstration. If you have obtained such numbers of cancer cells, why not just count them?

What’s more, this manuscript is lack of lots of key experiments to validate the proposed biosensor. For a biosensor, sensitivity, accuracy, and reproducibility are essential parameters the authors must analyze and discuss. The authors also need to provide data to support the experiment parameter, like why choose this frequency, why use this electrode width, spacing and so on.  I would also like to emphasize that the authors still provided no solid experimental evidence to support the conclusion of detecting different cells, especially for selectivity. For example, how do you distinguish CE81T-1/VGH and OE21-1 using photocurrent? How do you distinguish CE81T-1/VGH and OE21-1/VGH? Their calibration curves are so close. And at around 2500-3000 cells, the responses are quite similar. How do you detect these cells? Not to mention applied the proposed method in real samples.

The authors presented two methods. However, each method was not well discussed in depth.  There’s no benefit of combining the two methods together, nor the single method showed advantages. And there lacks a section to compare the proposed work to the literature.